# Methane Emission Reduction Enhanced by Hydrophobic Biochar-Modified Soil Cover

**Beibei Wu** [1,2], **Beidou Xi** [1,3,4], **Xiaosong He** [3,4], **Xiaojie Sun** [1,2,*], **Qian Li** [1,2], **Quanyi Ouche** [1,2], **Hongxia Zhang** [1,2] and **Chennan Xue** [1,2]

[1]   Guangxi Key Laboratory of Environmental Pollution Control Theory and Technology, Guilin University of Technology, Guilin 541004, China; b2946583125@163.com (B.W.); xibeidou@263.net (B.X.); q1643136880@163.com (Q.L.); s3140206113@163.com (Q.O.); mbyd0789@163.com (H.Z.); m18677363392@163.com (C.X.)

[2]   Guangxi Collaborative Innovation Center for Water Pollution Control and Water Safety in Karst Area, Guilin University of Technology, Guilin 541004, China

[3]   State Key Laboratory of Environmental Criteria and Risk Assessment, Chinese Research Academy of Environmental Sciences, Beijing 100012, China; hexs@craes.org.cn

[4]   State Environmental Protection Key Laboratory of Simulation and Control of Groundwater Pollution, Chinese Research Academy of Environmental Sciences, Beijing 100012, China

*   Correspondence: sunxiaojie@glut.edu.cn; Tel.: +86-150-7832-9789

**Abstract:** The microbial oxidation of $CH_4$ in biochar-modified soil cover is considered a potent option for the mitigation of emissions from old landfills or sites containing wastes full of low $CH_4$ generation rates. The mechanism of methane oxidizing bacteria (MOB) can be enhanced by amending the landfill cover soil with biochar, which is recalcitrant to biological degradation and can adsorb $CH_4$ while facilitating the growth and activity of MOB within its porous structure. However, the increase in the permeability coefficient and water content of the cover due to the addition of biochar also affects the methane removal efficiency. A hydrophobic biochar modified by KH-570 was employed to reduce the water content and to promote the diffusion and oxidation of $CH_4$ in the cover. Several series of small-scale column tests were conducted to quantify the $CH_4$ oxidation properties of the landfill cover soil amended with biochar and hydrophobic biochar under different levels of exposed $CH_4$ concentrations (5% and 15%), heights (10–66 cm), and temperatures (15–40 °C). After 30 days of domestication, the removal rate of the hydrophobic biochar-modified soil cover reached 98.8%. The water holding capacity of the cover and the $CH_4$ oxidation efficiency under different moisture contents were investigated in different columns. The hydrophobic biochar-modified soil cover has a weak water holding capacity, low saturated water content, and optimal $CH_4$ oxidation efficiency at this time.

**Keywords:** hydrophobic biochar; landfill cover; biocover; methane

## 1. Introduction

The presence of $CH_4$ is becoming a major problem and landfills are the largest anthropogenic methane sources worldwide [1]. $CH_4$ is considered to be a very powerful greenhouse gas (GHG) with a Global Warming Potential (GWP) of 28 over a 100-year time scale when compared to $CO_2$ [2]. Moreover, $CH_4$ has a lifetime of about 12 years in the atmosphere. Achieving landfill $CH_4$ emission reduction has important implications for global greenhouse effect control [3,4].

Nowadays, technologies to reduce $CH_4$ emissions in landfills can be divided into three categories: namely, resource utilization, end control, and in situ emission reduction. These categories have obvious limitations in practical applications. The resource utilization and end control of methane such as

heating, power generation, and torch burning could be utilized only in the active stage of landfill waste degradation. Studies have reported that in situ abatement technologies, such as quasi-aerobic landfills, can reduce $CH_4$ gas emissions for small and medium-sized landfills [5] However, the above-mentioned methods are not applicable when used in old or abandoned landfills with $CH_4$ concentrations below 20% [6,7]. Therefore, in the active stage of landfill waste degradation, combining engineering techniques and soil cover collection systems or using soil cover alone can significantly reduce $CH_4$ emissions in old and abandoned landfills through $CH_4$ adsorption and biochemical oxidation processes [8]. Scheutz et al. [9] reported that, in a landfill without a gas collection system, the ability of the soil cover to promote $CH_4$ oxidation is only 14%.

Biological cover that has been recommended by the authors of [10] can solve the above-mentioned problems. It optimizes the environmental conditions of the cover, enhances the $CH_4$ oxidation rates, and reduces the $CH_4$ fluxes emitted from landfills [11–13]. Biochar is a porous, organic material produced by the pyrolysis or gasification of waste biomass, and the increased presence of micropores makes it highly preferable for gas adsorption purposes [14]. China is one of the countries with the most abundant straw production in the world. According to surveys and statistics, the annual amount of straw in China from 2007 to 2009 was 735 million tons, and the conversion rate of pyrolysis to biochar was 35%. Compared with the environmental impact of burning straw in the open air, the total amount of biogas produced during the conversion of rice straw into biochar each year can provide a value of $7.27 \times 10^9$ kW · h for electricity production, which can reduce the $CO_2$ emissions from coal combustion by $7.78 \times 10^6$ t. Therefore, biochar is an economical and environmentally friendly material [15]. Hilger and Humer [16], Yargicoglu et al. [17], and Sadasivam and Reddy [18,19] indicated that $CH_4$ oxidation can be seen in landfill cover soil amended with biochar. Yaghoubi [1] also reported that the soil mixture permeability coefficient increases as the biochar content increases. This increment is beneficial to the diffusion and transportation of $O_2$ and $CH_4$. The aforementioned coefficient then expands the methane oxidation layer of the cover, increases the microbial activity and microbial density, and improves the $CH_4$ oxidation efficiency.

However, the increase of the permeability coefficient caused by the addition of biochar in the cover also promotes the diffusion and transportation of rainwater, thus increasing the production of leachate and affecting the oxidation efficiency of $CH_4$ [16]. Therefore, the biochar is hydrophobically modified and added to the cover to carry out a blanket methane oxidation test. This step is conducted to promote the diffusion of methane and oxygen and to prevent rainwater from entering the cover and the landfill [20].

The current study aims to investigate and compare the $CH_4$ transport properties of rice biochar and hydrophobic rice biochar at different exposed $CH_4$ concentrations, heights, and temperature values.

## 2. Materials and Methods

### 2.1. Biochar Pretreatment

The biochar used in the test is a solid substance containing a carbon element produced by waste rice straw in a completely anoxic environment through pyrolysis at 500 °C. Table 1 summarizes the physical and chemical properties of the biochar. The biochar used in the test is alkaline, with a high C content and low P and K content. The main component of rice straw is cellulose, and its main structures are sieve tube and conduit, which are the main reasons that rice straw biochar has a large specific surface area. The biochar was sieved through a no. 40–60 sieve mesh biochar and was placed in an electric blast drying oven at 150 °C for 24 h to thoroughly remove the original moisture and microorganisms in the biochar.

**Table 1.** Biochar physical and chemical indicators.

| Indicator | pH | C (%) | P (%) | K (%) | Ash (%) | Fill Density (g/cm$^3$) | Specific Surface Area (m$^2$/g) |
|---|---|---|---|---|---|---|---|
| Biochar | 10.8 | 64.2 | 0.16 | 0.33 | 30.2 | 0.131 | 66 |

### 2.2. Hydrophobic Modification of Biochar

The chemical hydrophobic modifier used as the biochar was silane coupling agent KH-570, which has a chemical name of 3-(methacryloyloxy) propyltrimethoxysilane. This agent is a colorless or yellowish transparent liquid commonly used for the surface treatment of inorganic fillers, such as white ash, talc, and clay. Figure 1 shows the surface hydrophobic modification reaction mechanism of KH-570. First, KH-570 undergoes a hydrolytic reaction, dehydrates, and then condenses to form an oligomer. Accordingly, the oligomer dehydrates with a hydroxyl group on the biochar surface to form a partial covalent bond, thereby coating the coupling agent on the biochar surface and increasing the hydrophobicity of the biochar surface.

**Figure 1.** Surface hydrophobic modification reaction mechanism of KH-570.

The pretreated biochar was immersed in a mixture of 20% (volume-to-volume ratio (v/v)) KH-570, 72% (*v/v*) absolute ethanol, and 8% (*v/v*) water in a volume percentage. The mixture was then stirred with a magnetic stirrer at 30 °C, after the biochar was thoroughly mixed with the modifier for 12 h. The modified biochar was rinsed with absolute ethanol three times and was filtered. The filtered sample was placed in a 50 °C electric blast drying oven for 4 h. In order to compare the hydrophobic properties before and after the biochar modification, we took 2 g of modified biochar into a glass funnel and covered it with a layer of water-soaked gauze to prevent the biochar from floating on the surface of the water. Then 50 mL of water was poured into the funnel. When the funnel no longer dripped water, the water absorption rate could be calculated by the weight of the dripped water. The water absorption rate was calculated as follows: (50-weight of dripping water)/2 × 100%. The lower the water absorption, the better the hydrophobic performance. Through experiments and calculations, we found that the water absorption of the modified biochar was 127%, which was much lower than the water absorption of 639% before modification.

### 2.3. Construction of Columns and Material Properties

Three columns were constructed from the polyvinyl chloride pipes with a length of 1000 mm and an inner diameter of 150 mm. An inlet for synthetic landfill gas was placed at the bottom, and an inlet for air was installed at the top. The top 100 mm of each column served as the air-filled headspace. A cover with a length of 700 mm was located between the air and the gravel layer that was placed at the bottom of each column with a length of 100 mm. Nine gas sampling points were mounted in 7 cm intervals, consisting of a tightly sealed butyl-rubber stopper and a separate sampling port was installed to the air outlet. On the back of the sampling port, three temperature sensors were installed, each spaced 20 cm apart to display the temperature conditions at different depths in the column. At the bottom of the pillar, a switchable water outlet was installed to provide drainage in case of leachate build-up. Figure 2 shows a schematic image of the setup.

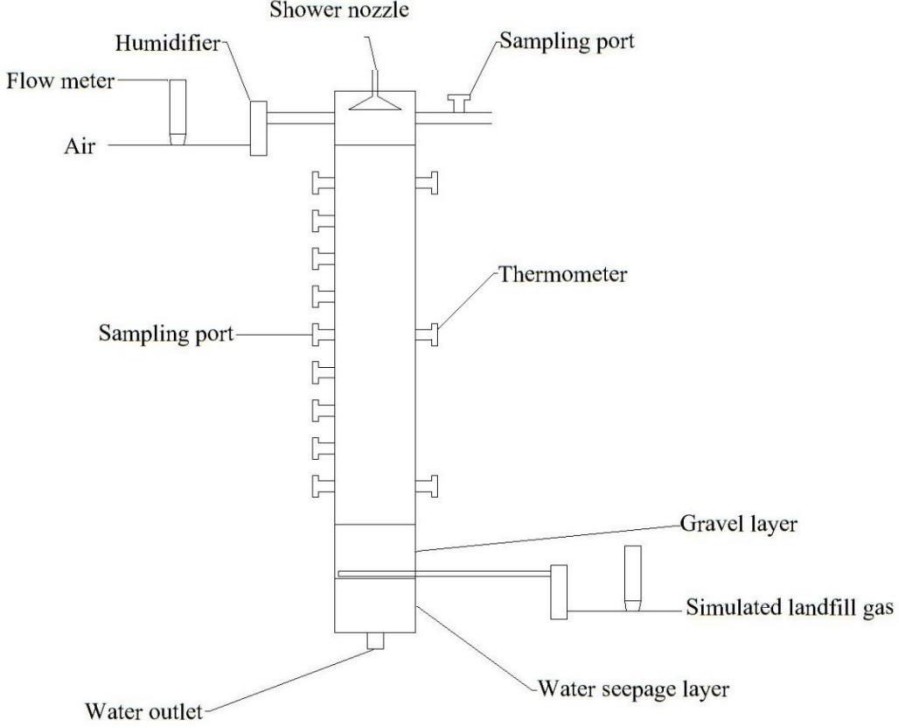

**Figure 2.** Schematic setup of the column experiment.

The landfill cover soil used for the laboratory experiments was a silty clay soil obtained from the Guilin Shankou Landfill. The soil was passed through a no. 10 sieve before being used in the batch or column incubation experiments. This step was carried out to produce a uniform soil texture throughout the columns and to minimize the formation of preferential flow paths, which can dramatically affect the gas retention time and oxidation rates in the columns. A column (Col. 1) was filled with pure soil. The biochar and soil, the hydrophobic biochar, and the soil were uniformly mixed at a volume ratio of 1:5 to obtain 20% *(v/v)* biochar-modified soil (Col. 2) and 20% *(v/v)* hydrophobic biochar-modified soil (Col. 3). Table 2 shows the characteristics of the covering materials. The physical and chemical characterization tests were performed in accordance with the standard methods. The columns were continuously charged with moisturized synthetic landfill gas (5% $CH_4$, 5% $CO_2$, and 90% $N_2$) at the flow rates of 13–15 mL/min. The inlet air rate of the upper part of the reactor was 50 mL/min, so as to simulate the natural atmospheric flow. Each column had a compartment connected to a heatable water tank to achieve water bath circulation, keeping the column temperature at about 25 °C. The gas samples were collected in a 100 mL gas collection bag. The $CH_4$ in the sample gas was analyzed by a gas chromatograph and was tested with a standard gas before each test to ensure the accuracy of the

test result. Each new flux was adjusted at least three days before the data collection commenced. This time was sufficient to exchange the whole gas volume in the columns at least once.

**Table 2.** Characteristics of three covering materials installed in the columns.

| Parameters | Col. 1 | Col. 2 | Col. 3 |
|---|---|---|---|
| pH | 5.57 | 8.18 | 7.46 |
| Compacted maximum dry density (g/cm$^{-3}$) | 1.94 | 1.56 | 1.58 |
| Optimum moisture content (%) | 18.1 | 34 | 32 |
| Plastic limit (%) | 17 | 32 | 36 |
| Liquid limit (%) | 27 | 53 | 45 |
| Plasticity index | 10 | 21 | 9 |
| P (%) | 0.08 | 0.1 | 0.09 |
| K (%) | 0.15 | 0.19 | 0.21 |
| Organic matter (%) | 4.1 | 14.75 | 14.93 |
| Moisture content (%) | | 10 | |

## 3. Results and discussion

### 3.1. $CH_4$ Oxidation in the Columns

During each phase, a stable vertical gas profile could be determined. The composition of the soil gas phase shows a high degree of variability among the different influx $CH_4$ concentrations. Figure 3 shows the $CH_4$ concentrations of three columns with time when the influx $CH_4$ concentrations were 5% and 15%. In the first 30 days, the column was operated at a concentration of 5% $CH_4$. As the actual $CH_4$ emissions will change according to environmental changes, we set a gradient change from 5% to 15% for the methane intake concentration. When the microorganism's oxidative capacity to 5% methane was stabilized, we increased the $CH_4$ intake concentration to 15% and continued to study its methane oxidation capacity. On day 30, the $CH_4$ intake concentration was adjusted to 15%. During the experiment, the $CH_4$ concentrations of the three columns all showed a downward trend. When the influx $CH_4$ concentration was 5%, Col. 2 had the lowest $CH_4$ concentration and Col. 3 had the highest one in the beginning. The increased levels of phosphorus, potassium, and organic matter in the soil are conducive to the growth and reproduction of $CH_4$ oxidizing bacteria due to the biochar addition; biochar also has an excellent adsorption effect [17–19]. However, the residual toxicity of the modifier in the biochar may affect the activity of the methane oxidizing bacteria (MOB). The rate of decrease in the $CH_4$ concentration of Col. 3 became fast as when the reactor was operated and $CH_4$ concentration of Col. 3 was lower than Col. 1 after 15 days. On day 30, the $CH_4$ in Col. 2 and Col. 3 was almost completely oxidized. The outlet $CH_4$ concentrations in Col. 2 and Col. 3 were 0.05% and 0.06%, and the removal rates were 99% and 98.8%, respectively. These findings indicate that the domestication of MOB over a period of time has gradually adapted to the new environment. Similar results were observed by Sadasivam and Reddy [18,19], that the $CH_4$ oxidation efficiency would be highly shown in landfill cover soil amended with biochar. In addition, landfill cover soil amended with hydrophobic biochar also has the same good $CH_4$ oxidation capacity.

When the influx $CH_4$ concentration was 15%, the $CH_4$ concentration of Col. 1 decreased the fastest in the first 10 days, followed by those in Col. 2 and Col. 3. However, the $CH_4$ concentration drop of Col. 3 maintained a relatively fast speed after 10 days. This occurrence indicated that the MOB inhibition by the toxicity of the residual modifier gradually decreased. At day 30, the outlet $CH_4$ concentrations of Col. 1, Col. 2, and Col. 3 were 6.59%, 3.12%, and 4.74%, respectively. Overall, Col. 2 had the best $CH_4$ oxidation efficiency, followed by Col. 3 with $CH_4$ removal rates of 79.2% and 68.4%, respectively. Hydrophobic biochar is more waterproof and has better breathability than biochar, and it is more beneficial to the growth and reproduction of methane oxidizing bacteria. The $CH_4$ concentrations increased accordingly with the $CH_4$ influx concentration in all columns, which was similar to the results of Yaghoubi [1]. At the same time, according to the trend prediction of the

decline rate of the CH$_4$ concentration after 30 days, the CH$_4$ oxidation efficiency of the soil modified by the hydrophobic biochar will exceed that of the biochar.

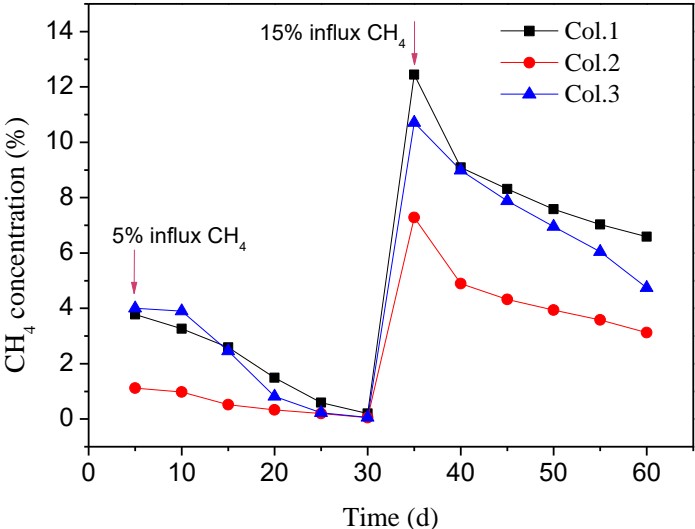

**Figure 3.** CH$_4$ concentration changes over time in the columns.

### 3.2. Effect of Temperature in the Columns

Temperature can significantly affect many microbial processes. Einola et al. [21] studied the CH$_4$ oxidation of a landfill in a cold zone and found that the optimum temperature was 19 °C. However, some studies have reported that the optimum temperature for methane oxidation is 20–30 °C [9,22,23]. We set a temperature group every 5 °C from 15 to 40 °C, a total of six different temperature test groups, and the other operating conditions were the same as above. Samples were taken at the same height outlet. The influx CH$_4$ concentration was 15%. The columns were run from a low to high temperature gradient and were incubated at each temperature for five days. Figure 4 shows that the outlet CH$_4$ concentration of each column gradually decreased as the temperature increased before reaching 30 °C. The outlet CH$_4$ concentration in Col. 1 was reduced to a minimum of 5.55% at 30 °C and the removal rate was 63%. When the temperature was 35 °C, the outlet CH$_4$ concentrations in Col. 2 and Col. 3 reached the minimum of 3.41% and 4.26%, and the removal rates were 77.27% and 71.6%, respectively. When the temperature was higher than 35 °C, the oxidation reaction of MOB was not as active as before. We assumed that the structure of the biochar and the additional nutrients increased the suitable temperature of the CH$_4$ oxidizing bacteria compared to with the soil. In summary, the optimum CH$_4$ oxidation temperature of the soil cover was found to be 30 °C. This finding is similar to the result of Börjesson et al. [22]. The optimum oxidation temperature of CH$_4$ for the 20% biochar-modified soil cover and the 20% hydrophobic biochar-modified soil cover is 35 °C. We assumed that the physical and chemical properties changed when the biochar was added to the soil. Consequently, the optimal temperature of the microorganisms in the biochar cover was slightly increased.

### 3.3. Effect of Depth in the Columns

The concentrations of CH$_4$ at different depths of the columns were measured continuously. The operating temperature of the columns were maintained at 25 °C and the operating conditions such as the moisture content were unchanged. The influx CH$_4$ concentration was 15%. The air outlet at the top of each column was used as the starting point. In addition to taking a sample at the air outlet, 10 gas samples were evenly spaced between 10 and 50 cm. Figure 5 shows the variation of CH$_4$ concentration with depth in three simulated reaction columns. The sampling ports on the column were numbered from zero to nine from top to bottom. The interval between the sampling ports was 7 cm. As can be seen in the figure, the concentration of CH$_4$ decreased and its oxidation efficiency increased as

the column depth decreased. Yaghoubi et al. [24] also obtained that the concentration profile along the column depth suggested that reactions were occurring predominantly in the top portions of the columns. At the same depth, the $CH_4$ oxidation efficiencies of Col. 2 and Col. 3 were slightly different but significantly improved compared with that of Col. 1. Most MOB are aerobic microorganisms. The $CH_4$ oxidation efficiency is rapidly increased compared with that obtained when the depth is below 40 cm. However, the $CH_4$ oxidation efficiency was enhanced in Col. 2 and Col. 3 compared with the soil cover.

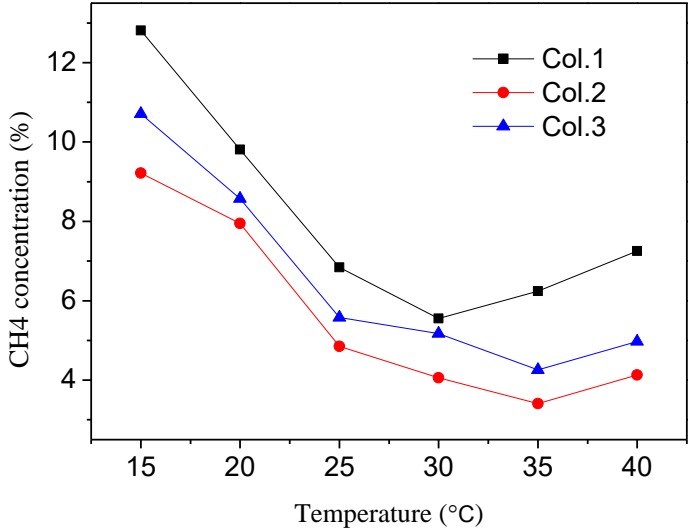

**Figure 4.** $CH_4$ concentration changes over temperature in columns.

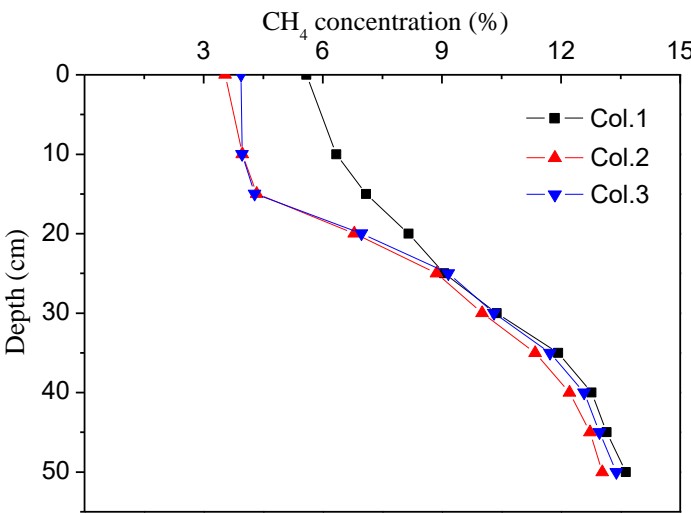

**Figure 5.** $CH_4$ concentration profile along the column depth.

*3.4. Effect of Moisture Content*

The effect of moisture on the $CH_4$ oxidation efficiency in the columns was investigated by adding an equal amount of water into the three columns and then measuring the changes in concentration profiles (Figure 6). The operating temperatures of the columns were maintained at 25 °C and the other operating conditions were the same as above. The samples were taken at the same height outlet. The influx $CH_4$ concentration was 15%. The initial moisture content of three columns were both 10%. We took a sample every hour because water began to contact the cover. Col. 2 had the highest moisture content when water was just added, followed by Col. 3. The biochar and hydrophobic biochar had a higher porosity than the soil. Thus, the carbonaceous cover had a high moisture content when the

cover was saturated with water. Kumar et al. [25] discovered a similar result. Moisture contents within the range of 10% to 20% gravimetric water content were reported as optimal for $CH_4$ oxidation in landfill cover soils [26,27]. However, Reddy et al. also indicated that biochar helps maintain adequate moisture throughout dry periods and to buffer the community against rapid changes in the moisture regime [19]. The high $CH_4$ content is due to the excessive moisture content that prevents oxygen from diffusing into the cover. The diffusion coefficient will be reduced by an order of magnitude when the water saturation is >85%, and the available $CH_4$ and $O_2$ in the liquid phase will also be drastically reduced, resulting in a decrease in oxidation efficiency [9]. In our results, the $CH_4$ content in each column began to decrease with the increase of time. The water content of Col. 3 rapidly decreased, and the final moisture content was low. This shows that the drainage performance of the hydrophobic biochar-modified soil cover was better in that it could reduce the inhibition of $CH_4$ oxidation by the high water content. The $CH_4$ oxidation capacity quickly recovered because the moisture in the pores of the cover was quickly discharged. Therefore, the $CH_4$ concentration in Col. 3 was also low after 6 h.

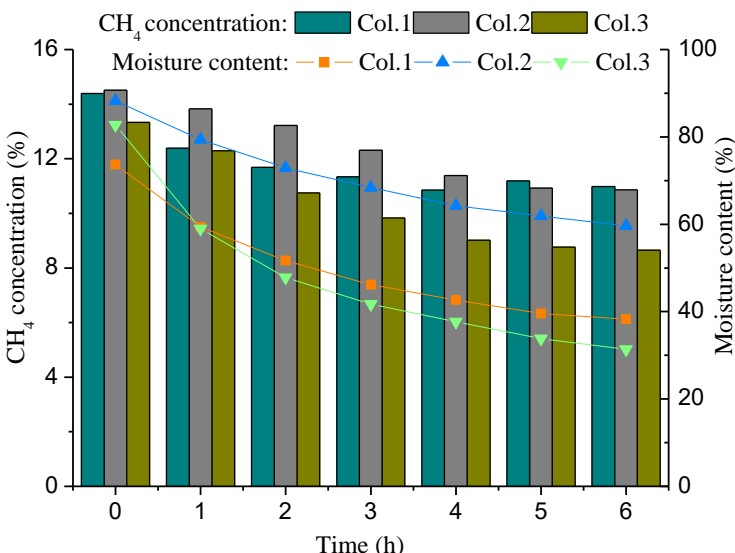

**Figure 6.** $CH_4$ concentration changes over the moisture content in columns.

## 4. Conclusions and Practical Implications

Landfill gas control is essential in preventing pollution and controlling global warming. The obtained data demonstrate that the methane removal efficiency from the soil cover with 20% biochar and hydrophobic biochar is much greater than that obtained from using the traditional soil cover. In this work, the chemical and physical characteristics and geotechnical properties of ordinary soil, biochar-amended soil, and hydrophobic biochar-amended soil were determined. The results showed that biochar amendment increased the pH level, phosphorus, potassium, and organic matter content of the soil, which led to a favorable soil environment for microorganism (methanotroph) growth and activities. This situation contributed to $CH_4$ mitigation from the landfill cover soil. The porous properties also enhanced the ability of the cover to adsorb $CH_4$. The modified hydrophobic biochar not only maintained the same characteristics as the biochar, but also improved the disadvantage of the high permeability of the biochar. Moreover, the modified hydrophobic biochar promoted the diffusion of $CH_4$ and oxygen in the landfill cover and helped reduce the source of leachate treatment in landfills. Accordingly, the $CH_4$ oxidation was accelerated and the $CH_4$ emission was reduced in old landfills. However, the leachate caused by the infiltration of rainwater into the cover was also reduced. During the simulated precipitation experiment, the water content of the hydrophobic biochar-modified soil cover rapidly decreased, and the final moisture content was low.

Finally, the temperatures and landfill depth wielded a huge impact on the methane oxidation efficiency of the hydrophobically modified biochar. Similar to the biochar-amended soil cover, the oxidation reaction of the MOB in the hydrophobically biochar-amended soil cover was less active at temperatures below 35 °C. Therefore, the optimum oxidation temperature of $CH_4$ for the hydrophobic biochar-modified soil cover was 35 °C. When the depth was <40 cm, the $CH_4$ oxidation effect increased as the depth decreased. However, the $CH_4$ oxidation efficiency was low when the depth was >40 cm.

**Author Contributions:** Conceptualization, B.W.; Data curation, B.W. and C.X.; Funding acquisition, X.S.; Investigation, B.X., X.H. and H.Z.; Methodology, C.X.; Project administration, B.X. and X.H.; Software, B.W. and Q.O.; Supervision, X.S.; Writing—original draft, B.W.; Writing—review and editing, X.S. and Q.L. All authors have read and agreed to the published version of the manuscript.

**Funding:** This research was funded by National Natural Science Foundation of China (No. 51668014), Natural Science Foundation of Guangxi (No. 2018GXNSFGA281001, 2018GXNSFAA138202), Science and Technology Major Project of Guangxi (GuikeAA18118013), and the Guangxi Science and Technology Planning Project (No. GuiKe-AD18126018).

**Conflicts of Interest:** The authors declare no conflict of interest.

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
