# Peer review of "Methane Emission Reduction Enhanced by Hydrophobic Biochar-Modified Soil Cover"

_processes, doi:10.3390/pr8020162_

Round 1

Reviewer 1 Report

The manuscript is good for publication in the current state. 

Author Response

Thank you very much for taking the time to review the manuscript for us. Attachment is our english editing certificate.

Reviewer 2 Report

The paper presents an interesting study about the application of biochar a soil cover. The manuscript is complete but the text is quite confusign and difficult to read; in addition, some aspects should be addressed before publication consideration:

English leguage must be revised and improved

Tables design cna be improved for more clarity

Sbet determination does not have enough accuracy to give results in decimal position. The Sbet of biochar must be expressed as 66 m2/g

No results about biochar yield are given by the authors. Thinking in a real application of this material as soil cover, this information becomes important

In line 90 a pretreatment of the biochar is mentioned but it is not explained in the manuscript; it is only exposed the biochar preparation by pyrolysis and KH570 treatment.

Point 2.3 Construction of columns is a bit confusing. A more clear and comprehensive wording of this part of the manuscript is needed

At line 118, the sythetic gas composition is given (5% CH4), but after 30 days operation, this concentration is changed to 15%; there is no explanation about the reason of this concentration variation. Also, authors should give an explanation about if any differences can be expected if 15%CH4 is fed to the columns from the first day instead of after 30d of operation

Series in Figure 3 should indicate the %CH4 at time = 0.

The conditions for variables influence studies (temperature, depht, moisture) are absolutely unclear. CH4 concentration in initial gas inlet and operating time must be clarified. Why the same conditions of previous part of the work are not used?

The biochar obtention is not cheap due to the energy consumption (also depending of the yiel; for this reason are important these results). The price of KH570 is high (around 7,000$/ton). And finnally, absolute ethanol is needed for the preparation procedure. So, I expect that the modified biochar has a very high cost, that could make non-viable this proposal. The authors should give an economic evaluation compared with alternatives used.

Reviewer 3 Report

The ms describe important subject related to methane emission from the soils increasing with global warming.

Although the subject is essential, there are many shortcomings in the publication that must be explained and corrected.

The introduction does not describe the results from other thematically similar works on the application of methane emission reduction enhanced by biochar in soil cover. Also, in the discussion, the results are not sufficiently quantitatively and qualitatively compared to similar ones already published. Consequently, one can have serious doubts as to the novelty of the presented results. The preparation, structure and properties of activated carbon have not been sufficiently well described, which, in consequence, does not make it possible to compare the results with others or to reproduce them. The results are not clearly described. Under the drawings, there is no clear legend making the figure self-explanatory. There is no appropriate statistical analysis. There is no clear comparison and how much better modified biochar is. The preparation of modified coal raises big doubts in terms of ecology, Life Cycle Assessment, industrial efficiency (cost) etc.

Round 2

Reviewer 2 Report

The authors have addressed all the points suggested and I think that now the manuscript can be published.

Reviewer 3 Report

In my opinion, in the present form, the ms is suitable for publication.